# Fabrication of CuO-NP-Doped PVDF Composites Based Electrospun Triboelectric Nanogenerators for Wearable and Biomedical Applications

**DOI:** 10.3390/polym15112442

**Published:** 2023-05-25

**Authors:** Bindhu Amrutha, Gajula Prasad, Ponnan Sathiyanathan, Mohammad Shamim Reza, Hongdoo Kim, Madhvesh Pathak, Arun Anand Prabu

**Affiliations:** 1Department of Chemistry, School of Advanced Sciences, Vellore Institute of Technology, Vellore 632014, India; 2School of Energy, Materials and Chemical Engineering, Korea University of Technology and Education, 1600, Cheonan-si 31253, Chungcheongnam-do, Republic of Korea; 3Department of Advanced Materials Engineering for Information & Electronics, College of Engineering, Kyung Hee University, Yongin-si 17104, Gyeonggi-do, Republic of Korea

**Keywords:** PVDF, CuO, electrospinning, triboelectric nanogenerator, health monitoring

## Abstract

A flexible and portable triboelectric nanogenerator (TENG) based on electrospun polyvinylidene fluoride (PVDF) doped with copper oxide (CuO) nanoparticles (NPs, 2, 4, 6, 8, and 10 wt.-% w.r.t. PVDF content) was fabricated. The structural and crystalline properties of the as-prepared PVDF-CuO composite membranes were characterized using SEM, FTIR, and XRD. To fabricate the TENG device, the PVDF-CuO was considered a tribo-negative film and the polyurethane (PU) a counter-positive film. The output voltage of the TENG was analyzed using a custom-made dynamic pressure setup, under a constant load of 1.0 kgf and 1.0 Hz frequency. The neat PVDF/PU showed only 1.7 V, which further increased up to 7.5 V when increasing the CuO contents from 2 to 8 wt.-%. A decrease in output voltage to 3.9 V was observed for 10 wt.-% CuO. Based on the above results, further measurements were carried out using the optimal sample (8 wt.-% CuO). Its output voltage performance was evaluated as a function of varying load (1 to 3 kgf) and frequency (0.1 to 1.0 Hz) conditions. Finally, the optimized device was demonstrated in real-time wearable sensor applications, such as human motion and health-monitoring applications (respiration and heart rate).

## 1. Introduction

Electricity shortages and the environmental issues associated with its generation have become a serious problem, due to the widespread usage of electronic devices, thereby impeding industrial and economic development [1,2,3,4,5]. Fossil fuels are used to generate 64% of worldwide electricity but, day by day, their availability is decreasing [6]. Numerous clean and viable energy sources have been researched to resolve this issue. Mechanical energy is one of the most prevalent sources of plentiful and easily available energy [2]. Many researchers have reported the development of nanogenerators (NGs) based on piezoelectric [6,7,8,9,10], pyroelectric [11,12,13], electromagnetic [14,15,16,17], and triboelectric [18,19,20] sensors for energy harvesting applications. Among these, piezo- and tribo-sensors have generated great interest among researchers, due to their flexibility, toughness, wide range of material options, and lightweight properties [21]. Owing to their mechanical energy-collecting properties, these sensors are efficient. They show promise for usage in health-monitoring sensors [22] and electronic devices [23]. Piezo sensors work under the principle of converting the dipole moment of a given mechanical energy into electrical energy [24]. Contrastingly, triboelectric NGs (TENGs) work through the shifting of electrons from one substance to another, with various charge affinities that come from the touching and subsequent separation actions [25,26].

Due to this simple operating principle, TENGs have received much interest, compared to other energy harvesting sensor devices [27]. In terms of how these devices operate, four major TENG modes have been developed: a free-standing mode [28,29], lateral-sliding mode [30], single-electrode mode [31], and vertical contact-separation mode [32]. The creation of tribo-charges at the interface or tribo-electrification, with changes in the distribution of the electric charge, or electrostatic induction are the different mechanisms that allow TENGs to harvest energy [33,34]. By applying an external pressure to a material surface, electrons will move from one surface to another, due to the difference in electron affinities [35]. Various triboelectric materials can be used to build a viable TENG device [36,37]. The two main characteristics that determine an object’s capacity to either gain or lose electrons are its polarity and surface-charge density [38]. Nylon, polyurethane (PU), and cotton can be used as tribo-positive materials. Polymers such as polyvinylidene fluoride (PVDF) and polyethylene terephthalate (PET) can act as tribo-negative materials [39,40].

Ferroelectric PVDF materials have attracted much attention due to their advantages, such as a high residual polarization, light weight, good mechanical properties, and high break-down voltage [41,42]. Semi-crystalline PVDF has five distinct crystalline phases (*α*, *β*, *γ*, *δ*, and *ε*) [43,44]. Of these, the *α*, *β,* and *γ*-crystalline phases are the most important. Usually, PVDF’s crystalline phase is dominated by the *α*-phase at ambient temperature conditions. The *α*-crystalline phase is non-polar with a trans-gauche structure and is an unstable phase. Because, the polymer chain crystallizes in an anti-parallel manner, resulting in non-electroactive *α*-phase. On the other hand, the *β* and *γ*-phases exhibit dipoles in a parallel orientation, are polar electroactive, and contain a net electric dipole moment [45,46]. The polar phase is exhibited by *β*-crystallinity [6,47,48]. In its all-trans conformation, PVDF exhibits high ferroelectric and piezoelectric properties, aided by the greater presence of the *β*-crystalline phase [49,50,51,52]. Researchers reported that PVDF pellets have 49.4% *β*-phase [52]. There are different methods developed for the preparation of PVDF films, such as solution-casting [53,54], 3-D printing [55,56], spin-coating [57,58,59,60], and electrospinning [61] methods. As-cast PVDF film shows around 60% polar *β*-phase [21]. Many researchers have focused on improving the *β*-phase in PVDF using the electrospinning method [62]; and in another report, PVDF electrospun at ambient temperature exhibited 83% *β*-phase [63].

Electrospinning is the most effective way of converting *α*-phase to *β*-phase, due to its unidirectional stretching and simultaneous poling effect [11,64]. It is a convenient and efficient method of generating high-specific-surface-area-based nanofibers [65]. Electrospinning techniques were re-popularized in the mid-1990s, due to increased studies into nanoscience [66,67,68]. There are many parameters that affect the nature of electrospun fibers, such as the nature of the polymer solution, electrospinning parameters, and environmental conditions. Of the polymer solution properties, the molecular weight, viscosity of the prepared solution, volatility of solvent, and solution conductivity are the most important parameters for determining fiber properties [69,70]. The applied voltage, distance between the collecting drum and needle tip, and rotating speed of the drum collector are the most important spinning parameters [71,72]. Finally, the humidity and temperature of the environment also affect the properties of electrospun fibers [73]. The resulting flexible fiber devices help to harvest energy effectively [74]. TENG output performance can be enhanced using various methods, such as using a single polymeric nanofiber, multi-layered NGs, surface modified electrospun fiber, and polymer nanocomposite fiber. The key to maximizing the electrical output of a tribo-based nanogenerator is material selection, including a wide range of electron affinities on the basis of the triboelectric series. A TENG uses the most tribo-positive and least tribo-negative materials, because this allows an exceptional electrical performance. A device’s effectiveness can be enhanced using physical or chemical modifications to the surface of the tribolayers. Additive and non-additive are the categories of physical modification. Lithography, laser printing, and plasma treatment are examples of physical non-additive enhancements. They greatly increase the frictional layer’s surface area and contact electrification, resulting in an excellent performance. Similarly, various dielectric additives, such as 0-D, 1-D, and 2-D nanoparticles, increase the triboelectric layer’s dielectric constants, which boosts the performance of the NG. In addition to physical alterations, TENG performance can also be improved through various chemical modifications such as spin-coating, electroplating, and electrospinning. In contrast to single modifications, hybrid improvements involving multiple physical or chemical changes produce improved outcomes. Of these, the most simple and effective doping of different materials, such as metals, metal-oxide, carbon-based materials, etc. into the PVDF matrix effectively enhances the output performance. Researchers evaluated the effect of conventional nanoparticles (NPs) such as Fe_3_O_4_, MWCNTs, MXene, graphene quantum dots, and ZnO nanowires on PVDF crystallinity [33,67,75,76,77,78,79]. Among them, CuO is a promising semiconductor that is gaining popularity, owing to its superior structural, electromagnetic, and optical properties [80]. It has a 1.2 eV band gap. This narrow band gap allows its application in solar cells, sensors, and catalytic reactions [81]. CuO nanostructures are synthesized in the form of nanowires, nanorods, and NPs [82]. Different methods for preparing NPs of different sizes and shapes have been suggested over the last few decades, including wet-chemical [83], thermal oxidation [84], co-precipitation [85], and sono-chemical [81] methods. Among these, the simplest process is the coprecipitation method, which is gaining popularity in industry, due to its low heating rate and power requirements, cost-efficient method for mass production, and higher yield [86].

The present study was primarily focused on improving the triboelectric performance of electrospun PVDF by doping with CuO NPs. Typically, materials from both the positive and negative sides of the triboelectric series are used to fabricate a TENG [87]. Here, a CuO-doped PVDF electrospun fiber material was used tribo-negative and a PU film was utilized as the counter-positive material for the tribosensor. As PVDF is a familiar piezoelectric material, we constructed the device in such a manner that only one PVDF surface was left linked to the electrode. Piezoelectricity occurs only when both fiber surfaces are in contact with the electrodes [88,89]. Thus, the influence of piezoelectricity in a TENG can therefore be avoided [90]. In particular, a NP-doped PVDF material successfully increased the yield of TENG, because of its better electronegativity and conductive nature. Additionally, the TENG was self-powered and linked to numerous body areas, to track a variety of human activities, and was demonstrated to have an excellent energy-harvesting capability.

## 2. Materials and Methods

### 2.1. Reagents

PVDF with an average molecular weight of M_W_ ~ 370,000 g·mol^−1^ was purchased from Solvay, Republic of Korea. Copper sulfate pentahydrate (CuSO_4_·H_2_O) and sodium hydroxide (NaOH; 98%) were supplied by Sigma-Aldrich (St. Louis, MO, USA). Acetone and dimethylformamide (DMF) were bought from Merck (Darmstadt, Germany). No further purification was performed before using the materials.

### 2.2. Synthesis of CuO NPs

CuO NPs were synthesized using a facile coprecipitation method. Precursors of CuSO_4_·5H_2_O diluted with distilled water (0.1 M solution) were used for the synthesis of CuO NPs. Then, the precursor solution was maintained at room temperature (R.T.) in a magnetic stirrer. Next, 0.1 M NaOH was dropwise added to the solution after 2 h of stirring, until the solution pH reached 14. The solution color changed to black, and the precipitate was centrifuged and washed several times with distilled water to maintain the pH at 7. The collected precipitate was then left to dry for 24 h at 80 °C. Finally, the dried precipitate was calcinated for 5 h at 500 °C [84,91], as shown in Figure 1a. The resulting NPs were characterized using Fourier transform-infrared spectroscopy (FTIR) and an X-ray diffractometer (XRD).

### 2.3. Preparation of CuO-Doped PVDF Electrospun Fiber

DMF and acetone were used in a 6:4 ratio to prepare a 12 wt.-% PVDF electrospinning solution. Initially, PVDF was dissolved in DMF and the mixture was agitated for 2 h at 60 °C. The above-prepared solution was then supplemented with CuO (2, 4, 6, 8, and 10 wt.-%) NPs and agitated for an additional 1 h at R.T. Meanwhile, acetone was added, to maintain the solution viscosity, and the mixture was stirred for a further 24 h at R.T. Electrospinning of neat PVDF and NP-doped PVDF was carried out using an electrospinning apparatus with a single jet (ESPIN NANO V1-VH). The setup consisted of a high voltage application source, 10 mL plastic syringe, and a stainless-steel needle of 23 G size. A voltage-supply source capable of producing a direct positive current was attached to the needle tip. The distance between the needle tip and collector was maintained at 10 cm and the applied positive voltage was fixed at 20 kV. The polymer solution was transferred through the syringe pump at a controlled flow rate of 1.0 mL/h [92]. A Teflon sheet was used to collect the electrospun fiber. The electrospinning process was completed at R.T. [47], as shown in Figure 1b.

### 2.4. Fabrication of the TENG Device

CuO-doped PVDF electrospun fiber was used as the counter-negative material and a PU film was utilized as the positive layer for fabrication of the triboelectric device. Both positive and negative tribo-layers were initially sliced into 2.0 × 2.0 cm^2^ pieces and adhered to the adhesive sides of the flexible Ni-Cu electrodes. Then, the two films were positioned directly across from one another. Finally, the entire device was enclosed in PET film, which helped to shield the device from ambient noise and enhance the output performance.

### 2.5. Characterization

The particle size and phase structure of the NPs were examined with D-8 advanced XRD with Cu-Kα radiation, using a 10°–90° range of 2ϴ. By using FE-SEM and EDS techniques, the surface morphology and composition of elements in the electrospun fiber were studied. The fiber was treated with gold sputtering before the SEM examination. The non-polar to polar phase change in PVDF was analyzed using FTIR and XRD techniques. A customized dynamic tester was used to measure the electrical output in the form of peak-to-peak voltage (*V*_p-p_), by applying a periodic load and frequency. BIOPAC MP 150 (BIOPAC Systems, Inc.: Goleta, CA, USA) data acquisition equipment with an integrated piezo amplifier was used for the health monitoring applications and acknowledgment 4.2 Software was used to wirelessly save the data in a PC.

## 3. Results

### 3.1. Characterization Studies

#### 3.1.1. Characterization of CuO NPs

The crystalline nature of the CuO NPs was confirmed from the XRD pattern (Figure 2a). The diffraction peaks observed at 32.46°, 35.6°, 38.71°, 48.67°, 53.77°, 58.37°, 61.52°, 66.37°, and 68.21° corresponded to the (110), (002), (111), (202), (020), (202), (113), (311), and (113) miller indices planes of crystalline peaks, and matched the JCPDS card no. 45-0937 [93]. This confirmed the monoclinic structure of the CuO NPs. The characteristics of the diffraction peaks associated with the monoclinic crystal structure of CuO were consistent with the peaks. Using Scherrer’s equation, as given in Equation (1) [94], the average crystalline size was calculated from the XRD peaks.
d = K λ/(*β* cosϴ)(1)
where K is the form factor with a value of 0.9; the wavelength of the used X-ray beam is represented as λ, which is about 1.54 Å; *β* represents the full width at half maximum (FWHM) of the diffraction peak; and ϴ is the diffraction angle. The average crystalline size of the CuO NP matching the maximum peak shown in XRD was measured to be 24 nm, and the synthesized NPs were highly pure. The CuO NPs were believed to be nanocrystalline, because the XRD pattern showed strong structural peaks and particles with sizes below 100 nm. The functional groups present in the CuO NPs were confirmed with FTIR characterization. Absorption bands at 608 cm^−1^ and 726 cm^−1^ represented Cu–O vibrational deformation. There was symmetric, as well as asymmetric, stretching of Cu–O at the peak of 1646 cm^−1^. The 1060 and 1119 cm^−1^ peaks were assigned to the C–O plane, as shown in Figure 2b. The peaks closely matched the literature values [95].

#### 3.1.2. Characterization Studies of the Neat and Doped PVDF Fibers

The functional groups of the electrospun neat PVDF and CuO-doped PVDF were analyzed using FTIR. The spectra of the PC–0, PC–2, PC–4, PC–6, PC–8, and PC–10 (refer to sample codes in Table 1) fibers were noted as (a) to (f), as shown in Figure 2c. The highly active vibration peaks for the fibers were found to be in the range of 500 to 1600 cm^−1^. The characteristic peak at 1400 cm^−1^ was assigned to –CH bending vibration. –CF_2_ stretching vibrations in the chemical structure occurred at 1180 cm^−1^. Extremely weak peaks at 607, 766, and 976 cm^−1^ [96,97] corresponded to the amorphous phase. The crystallinity and polar *β*-phase of the electrospun fibers were enhanced by the typical peaks at 840 and 1276 cm^−1^ [96,98,99].

The doping of CuO NPs resulted in a significant enhancement of polar phase, while the amorphous phase was decreased. Peaks at 766 and 840 cm^−1^ corresponding to both *α*-and *β*-phases were used to calculate the percentage increase in the *β*-phase content, as given in Equation (2) [100].
F(*β*) = [(A*_β_*)/((1.26 × A*_α_*) + A*_β_*)] × 100(2)
where A*_α_* and A*_β_* are the absorption peaks corresponding to 766 and 840 cm^−1^. A quantitative graph of *β*-phase content is shown in Figure 2d and Table 1. The polar phase peak strength was considerably enhanced when increasing the NP concentration from 2 to 8 wt.-% compared to neat PVDF. A further increase in the NP concentration to 10 wt.-% showed a decrease in *β*-phase percentage, possibly due to NP agglomeration in the polymer surface. The enhancement of the polar phase in the doped PVDF compared to the neat PVDF was confirmed using XRD analysis. Figure 2e depicts the XRD peaks of neat PVDF and CuO-NP-doped PVDF electrospun fibers. The fibers showed a significant peak of 20.4°, equivalent to the crystalline *β*-phase, and a very weak peak at 18°, corresponding to the presence of the α-crystalline peak, but its intensity decreased with increased NP doping. The CuO-doped fibers had two extra small peaks at 35.3° and 38.5°, confirming the doping of NPs in the PVDF material. The 10 wt.-% NP-doped fiber showed a merged peak instead of two different peaks, which may have been due to the agglomeration of NPs in the PVDF fiber.

Figure 3 shows the surface morphology and elemental analysis of the electrospun PVDF and CuO-NP-doped PVDF fiber using FE-SEM and EDS. Figure 3a,b shows the surface morphology of the neat fiber. Figure 3c,d shows the surface morphology of the PC–8 fibers. Figure 3e shows the EDS analysis of neat PVDF (PC–0). Figure 3f,g shows the EDS mapping and elemental analysis of 8 wt.-% CuO-doped PVDF fiber. PC–8 showed better polar phase properties compared to the other compositions. Therefore, surface analysis was carried out for the optimized PC–8 sample. The fiber structures showed better structural stability, and the PC–0 fiber diameter was 38–110 nm. The average diameter of the doped fiber ranged from 250 to 300 nm. All fibers were arranged randomly. The EDS elemental analysis confirmed the successful doping of CuO NPs into PVDF.

#### 3.1.3. Working Principle of TENGs

A contact separation method was used for studying the mechanism of the TENG device. A schematic view of the TENG device is shown in Figure 4a. Neat/doped PVDF fiber served as a negative material in this instance, while the PU film served as a positive layer. As demonstrated in step (i) (Figure 4b), the two layers displayed neutral charges before applying an external pressure. When an external load was applied, the PVDF layer made contact with the PU film. The triboelectric principle states that electrostatic induction will occur when two layers with different electron affinities come into contact with each other. The PU lost its electrons more quickly than the PVDF and it became positively charged, while the PVDF layer became negatively charged. Due to the electric balance in step (ii), current flow did not occur. The layers gradually separated from one another after releasing the external load, thereby creating a charge difference between the electrodes. In order to stabilize the potential variation, electrons started to travel from higher to lower concentration, as shown in step (iii). This continued till the layers and respective electrodes achieved the same number of opposite charges. due to electrostatic induction. As a result, the current flow occurred in step (iv). When we compressed further, it lost its electrical balance, which caused the flow of electrons in the opposite direction, as depicted in step (v) [101]. A constant flow of AC current was produced by a regular cyclic action of touching and splitting.

#### 3.1.4. Triboelectric Sensor Study

The triboelectric responses of the fabricated PVDF/PU and PVDF–CuO/PU composites are shown in Figure 4c. Before the experiments, the prepared fibers were enclosed in PET sheets, which helped to reduce noise and enhance the output of the device. By using a customized dynamic pressure system, the triboelectric performance of the composites was examined. The *V*_p-p_ of PVDF/PU and PVDF−CuO/PU were measured by applying a constant applied load of 1.0 kgf at 1.0 Hz frequency. For increasing CuO NP contents in the PVDF (0, 2, 4, 6, 8, and 10 wt.-%), the output voltages of the sensors were calculated to be 1.71, 2.05, 4.34, 6.2, 7.5, and 3.9 V, respectively, under an input impedance of 100 MΩ and 20 dB gain. The output voltage increased by approximately four times in the PC–8/PU composite (7.5 V) compared to the PC–0/PU (1.7 V). It was anticipated that the electrospinning technique caused the PVDF’s C−F dipoles to self-polarize, where a hydrogen atom from one PVDF unit aligned with another unit containing a highly electronegative fluorine atom by hydrogen bonding. The electrospinning process caused the electrification of the polymer solution under the strong applied electric field (20 KV). Additionally, the doping of CuO NPs caused the polarization of PVDF. Up to 8 wt.-% doping of CuO NPs into PVDF enhanced its electroactive *β*-crystalline phase. However, the reduced *β*-crystalline phase with the CuO NP doping beyond 8 wt.-% may have been due to the increased coagulation during sample preparation. In addition to becoming more viscous, the quality of the formed fiber was poor compared to the PC–8 fiber. It is evident from the FTIR and XRD measurements that the addition of 10 wt.-% CuO NPs to PVDF reduced the percentage of *β*-phase content, from 87% (for PC–8 fiber) to 79%.

Quantitative data of the voltages are given in Appendix A. Further measurements were carried out for the optimized sensor. The variation in output voltage relative to an applied load from 2 to 3 kgf under a constant frequency of 1 Hz, as well as varying the frequency from 0.1 to 0.5 Hz with a constant load of 1.0 kgf, is shown in Appendix A. A voltage study with 2 kgf and 3 kgf was carried out at 0 dB gain and 100 MΩ input impedance. We obtained 1.3 V under a 2 kgf load and 1.87 V under 3 kgf. The TENG produced 8.7 V and 6.3 V under 0.5 Hz and 0.1 Hz applied frequencies and 1 kgf load, respectively. The measurements were carried out at a 20 dB gain and 100 MΩ impedance.

#### 3.1.5. Wearable Applications

Finally, the TENG device was used to produce biomechanical energy from daily human motions and in health-monitoring applications, such as respiration and heartbeat monitoring. The sensor was used to monitor various human body movements. Figure 5 illustrates the different applications of the PC–8/PU triboelectric sensor performed at 100 MΩ and 0 Db gain. Figure 5a shows the output electrical voltage of the sensor while tapping, bending, twisting, and rolling. The optimized sensor generated 6.6 V at most with tapping, 16.3 V at most while twisting, 14.4 V while folding, and 4.9 V when rolled. These results suggest that the tribosensor responded better to the bending and twisting than to tapping and rolling.

The sensor could distinguish between slow (10 V) and fast (15.8 V) movements when fixed in the elbow; in this case, faster movements generated a higher voltage than slower movements, as depicted in Figure 5b. More interestingly, we used it as a theft monitor sensor, as shown in Figure 5c. Figure 5d–f demonstrate that the triboelectric sensor could distinguish between different body movements, such as walking, jumping, sitting, and kicking, when it was positioned in jeans, a chair, and in a shoe. To verify the practicability of the device’s function as a health monitoring sensor, such as a respiratory and heartbeat monitor, the sensor was placed under the chest in a prone position as shown in Figure 5g. It produced primary and secondary respiration peaks and primary heartbeat peaks at 0.33 Hz, 0.66 Hz, and 1.01 Hz, respectively. Based on these findings, the proposed PVDF−CuO (PC–8)/PU tribosensor can successfully capture the different forms of biomechanical energy prevalent in everyday human activity.

## 4. Conclusions

In this work, a composite fiber made of a CuO−NP−doped PVDF fiber and PU film was used to create stretchable energy harvesting triboelectric sensors, in which the PU film and PVDF fiber were utilized as the positive and negative tribo-materials, respectively. The CuO NPs were synthesized through an effective coprecipitation method and characterized using FTIR and XRD. Through the electrospinning method, we successfully fabricated various percentages of CuO−doped PVDF (2, 4, 6, 8 and 10) fiber with an enhanced polar *β*-phase, and these were confirmed with XRD, FTIR, and SEM analyses. Finally, the triboelectric efficiency of the fabricated sensors was studied, and an impressive *V*_p-p_ of 7.5 V was achieved for PC–8, which is around 4.5-times higher than the 1.7 output voltage achieved for PC–0 under an optimized load of 1 kgf and frequency of 1 Hz. The PVDF−CuO (PC–8)/PU triboelectric sensor was effectively used as an energy harvester and could be used to detect various body movements (finger, leg, and elbow). We obtained encouraging results, which demonstrated a potential application of the sensor in smart mattress for medical purposes, particularly for monitoring patients in the coma stage. In addition, it can measure a person’s heart rate, blood pressure, and respiration rate; hence, this sensor device has the potential for use in clinical trials.

## Figures and Tables

**Figure 1 polymers-15-02442-f001:**
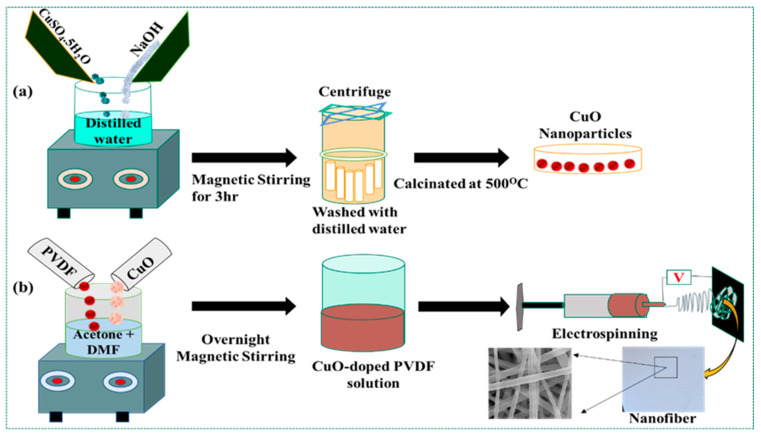
(**a**) Schematic representation of the preparation of CuO NPs; (**b**) schematic representation of CuO-doped PVDF electrospun fiber formation.

**Figure 2 polymers-15-02442-f002:**
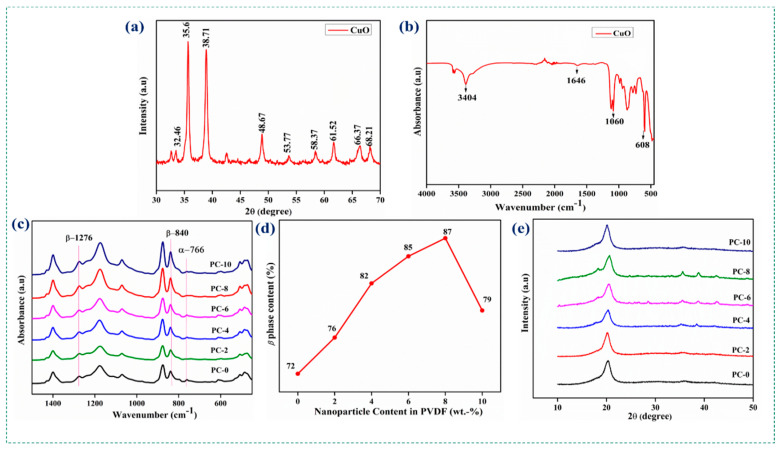
(**a**) XRD pattern and (**b**) FTIR spectrum of CuO NPs; (**c**) FTIR spectra and (**d**) quantitative graph of *β*-phase content (%); (**e**) XRD pattern of electrospun neat PVDF and 2, 4, 6, 8, and 10 wt.-% CuO NP-doped PVDF.

**Figure 3 polymers-15-02442-f003:**
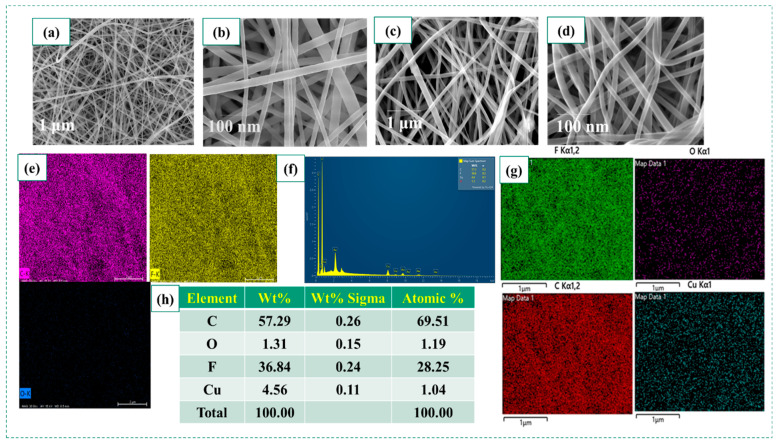
SEM images of (**a**,**b**) PC–0; (**c**,**d**) PC–8; EDS analysis of (**e**) PC–0; (**f**,**g**) PC–8; (**h**) elemental mapping of PC–8.

**Figure 4 polymers-15-02442-f004:**
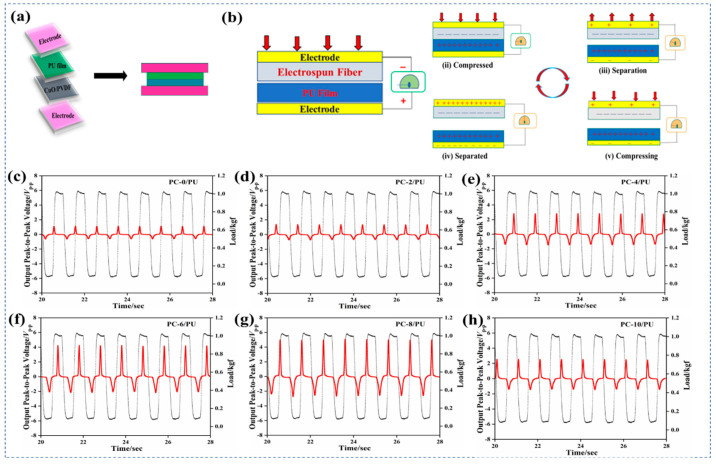
(**a**) Graphical view of the TENG device; (**b**) diagram illustrating the charge generating process of the PVDF–CuO TENG; (**c**–**h**) Peak-to-peak output voltage of the neat PVDF and PVDF–CuO (2, 4, 6, 8, and 10 wt.-%)/PU TENGs.

**Figure 5 polymers-15-02442-f005:**
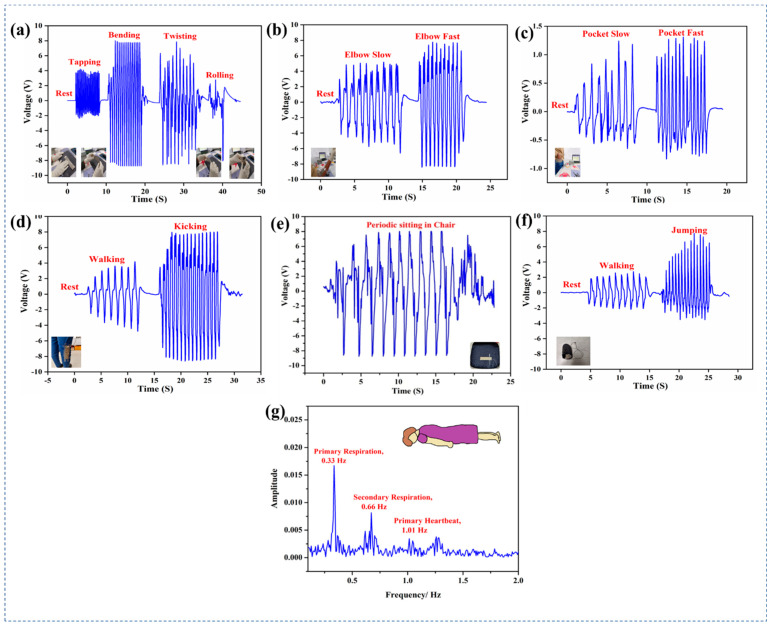
(**a**) Wearable applications of PC–8/PU TENGs sensing performance while tapping, bending, twisting, and rolling; (**b**) sensing performance of elbow movements; (**c**) theft monitor sensor; (**d**) sensor performance for leg movements; (**e**) applications for health care in smart chairs; (**f**) motion detection when sensor was connected to a shoe; (**g**) respiration and heartbeat monitoring of the human when the sensor is placed near to the chest during prone position.

**Table 1 polymers-15-02442-t001:** *β*-crystalline phase % of electrospun neat PVDF and PVDF doped with CuO NPs (2, 4, 6, 8, and 10 wt.-%).

Sample Code	Composition (wt. Ratio)	Absorption Intensity		F(*β*) %
		A*_α_*	A*_β_*	
PC–0	PVDF: CuO (100:0)	0.0310	0.1036	72
PC–2	PVDF: CuO (98:2)	0.0723	0.2910	76
PC–4	PVDF: CuO (96:4)	0.0912	0.5064	82
PC–6	PVDF: CuO (94:6)	0.1120	0.7248	85
PC–8	PVDF: CuO (92:8)	0.1130	0.9770	87
PC–10	PVDF: CuO (90:10)	0.2480	1.2086	79

## Data Availability

The data will be made available on request.

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
