# Peer review of "Fabrication of CuO-NP-Doped PVDF Composites Based Electrospun Triboelectric Nanogenerators for Wearable and Biomedical Applications"

_polymers, 2023, doi:10.3390/polym15112442_

Round 1
Reviewer 1 Report
The manuscript created stretchable energy harvesting triboelectric sensors using CuO NPs doped PVDF fiber and PU film. Their research is very interesting. The rusults are abundant and the process is clear. I have no questions. I think the manuscript can be accepted now.
Reviewer 2 Report
The research group develops a triboelectric nanogenerator device with perspectives of applicability in the field of health. Characterization of the material was carried out with good techniques and the obtained results are presented in a correct way with a good review of the state of the art and the discussion is well supported in accord with fundamental and current bibliographical references.
- I recommend improving the quality of the images so that they can be well understood by readers.
- Also, from my point of view, the title of this research work does not describe or encompass the substantial content of the project. I suggest rewriting the title so that it better projects the research work.
- The approach proposed is interesting, but do you think that such an approach is really feasible in clinics?
Reviewer 3 Report
After reading your manuscript, I find it interesting to handle different formats of nanostructures to achieve a sensory effect that you have managed to reflect in a brilliant way. So I congratulate you for the work. I have a couple of suggestions.
-The first is that it is not well established the value of the size of nanoparticles, which would require electron microscopy (TEM) or dynamic light scattering (DLS). Or at least a suitable reference that specifies the size of the nanoparticles. Important data in order to reproduce their work and really explain the effects of the 10%wt CuO Nps.
-Second, regarding the format of the tables i) and J) into Figure 3, I would try to unify them to improve the presentation.
